# Transcriptome-Wide Characterization of piRNAs during the Developmental Process of European Honey-Bee Larval Guts

**DOI:** 10.3390/genes13101879

**Published:** 2022-10-17

**Authors:** Ya-Jing Xu, Qi Long, Xiao-Xue Fan, Ya-Ping Ye, Kai-Yao Zhang, Jia-Xin Zhang, Hao-Dong Zhao, Yu-Tong Yao, Zhong-Min Fu, Da-Fu Chen, Rui Guo, Ting Ji, Zhe-Guang Lin

**Affiliations:** 1College of Animal Sciences (College of Bee Science), Fujian Agriculture and Forestry University, Fuzhou 350002, China; 2Apitherapy Research Institute of Fujian Province, Fuzhou 350002, China; 3College of Animal Science and Technology, Yangzhou University, Yangzhou 225000, China

**Keywords:** honey-bee, *Apis mellifera ligustica*, piRNA, gene expression, regulatory network, larva, gut, development

## Abstract

piRNAs play pivotal roles in maintaining genome stability, regulating gene expression, and modulating development and immunity. However, there are few piRNA-associated studies on honey-bees, and the regulatory role of piRNAs in the development of bee guts is largely unknown. Here, the differential expression pattern of piRNAs during the developmental process of the European honey-bee (*Apis mellifera*) larval guts was analyzed, followed by investigation of the regulatory network and the potential function of differentially expressed piRNAs (DEpiRNAs) in regulating gut development. A total of 843 piRNAs were identified in the larval guts of *A. mellifera*; among these, 764 piRNAs were shared by 4- (Am4 group), 5- (Am5 group), and 6-day-old (Am6 group) larval guts, while 11, 67, and one, respectively, were unique. The first base of piRNAs in each group had a cytosine (C) bias. Additionally, 61 up-regulated and 17 down-regulated piRNAs were identified in the “Am4 vs. Am5” comparison group, further targeting 9, 983 genes, which were involved in 50 GO terms and 142 pathways, while two up-regulated and five down-regulated piRNAs were detected in the “Am5 vs. Am6” comparison group, further targeting 1, 936 genes, which were engaged in 41 functional terms and 101 pathways. piR-ame-742536 and piR-ame-856650 in the “Am4 vs. Am5” comparison group as well as piR-ame-592661 and piR-ame-31653 in the “Am5 vs. Am6” comparison group were found to link to the highest number of targets. Further analysis indicated that targets of DEpiRNAs in these two comparison groups putatively regulate seven development-associated signaling pathways, seven immune-associated pathways, and three energy metabolism pathways. Moreover, the expression trends of five randomly selected DEpiRNAs were verified based on stem-loop RT-PCR and RT-qPCR. These results were suggestive of the overall alteration of piRNAs during the larval developmental process and demonstrated that DEpiRNAs potentially modulate development-, immune-, and energy metabolism-associated pathways by regulating the expression of corresponding genes via target binding, further affecting the development of *A. mellifera* larval guts. Our data offer a novel insight into the development of bee larval guts and lay a basis for clarifying the underlying mechanisms.

## 1. Introduction

Piwi-interacting RNAs (piRNAs) are types of small non-coding RNAs (ncRNAs), with a length distribution from 24 nucleotide (nt) to 32 nt [1]. Accumulating evidence shows that piRNAs play critical roles in suppressing transposons and maintaining genome stability [2,3]. Dissimilar to miRNA, piRNAs are transcribed from single-stranded RNA via a dicer-independent mechanism and function by interacting with P-element-induced wimpy testis (Piwi) proteins [4,5]. Structurally, piRNAs have no secondary hairpin structures, with a uridine (U) base bias at the 5′ end and an adenosine (A) at position 10 [6]. However, piRNA has been proved to specifically bind to target mRNA in an miRNA-like way [7]. There are two major pathways to generate piRNAs: the primary processing pathway and the ping-pong cycle that amplifies secondary piRNAs [8]. In somatic and germline cells, piRNA precursors are transcribed from piRNA clusters in the nucleus, which are processed by the endonuclease Zucchini (Zuc) to produce piRNA intermediates, and subsequently piRNAs can be loaded into Piwi or Aubergine (Aub) to form piRNA-induced silencing complexes (piRISCs), which are further trimmed by exonuclease and 2′-O-methylation of piRNAs with Hen1 (nascent helix-loop-helix 1) to produce mature piRNAs. Whereas secondary piRNAs are produced only in germ cells by secondary expansion of Argonaute 3 (AGO3) protein with Aub-piRISCs. This process is also known as the ping-pong pathway [6,9,10,11]. Recent studies demonstrated that piRNAs, as newly emerging regulators of gene expression, could exert functions in an array of biological processes such as immune response and development [12,13]. For example, Wang et al. [14] discovered that piRNAs in *Aedes albopictus* were involved in the host antiviral immune response to Dengue virus 2 (DENV-2) infection. 

Honey-bees are capable of pollinating a substantial quantity of wildflowers and crops, thus playing a pivotal part in the maintenance of ecological balance and the survival of mankind [15]. As one of the most widely distributed subspecies of *A**. mellifera*, *A. m. ligustica* are commercially reared in China and other countries for their great economic value [16]. The insect gut is an essential organ for food digestion, nutrient absorption, and immune defense [17]. Previous works were mainly focused on the adult bee gut and intestinal microorganisms [18,19]. Previously, our group conducted a series of studies on ncRNA-regulated development of honey-bee guts, e.g., Guo et al. systematically identified long non-coding RNAs (lncRNAs) in *A. m. ligustica* workers’ midguts and analyzed DElncRNA-regulated parts in midgut development [20]. Guo et al. deciphered the differential expression profile of circle RNAs (circRNAs) in the midgut tissues of *A. m. ligustica* workers and putative roles of DEcircRNAs in midgut development [21]. Guo et al. also conducted transcriptome-wide identification of microRNAs (miRNAs) in the *A. m. ligustica* larval guts followed by investigation of DEmiRNA-mediated regulation of gut development [22]. However, little progress on piRNAs engaged in regulating the development of the bee gut has been made to date.

In the previous study, our group performed deep sequencing of 4-, 5-, and 6-day-old larval guts of *A. m. ligustica* followed by the identification and investigation of miRNAs using bioinformatics [23]. Currently, whether and how piRNAs regulate the development of *A. m. ligustica* larval guts are still largely unknown. Here, to decipher the expression profile of piRNAs during the developmental process of *A. m. ligustica* larval guts, piRNAs in larval guts were identified and validated based on the obtained high-quality sRNA-seq datasets, and differentially expressed piRNAs (DEpiRNAs) were then analyzed followed by target prediction and regulatory network investigation. DEpiRNAs and corresponding target genes that were associated with the development of the larval gut are discussed. the findings in the present study will provide a novel insight into the development of the honey-bee larval gut and a basis for illustration of the piRNA-regulated mechanism underlying gut development.

## 2. Materials and Methods

### 2.1. Bee Larvae

The *A. m. ligustica* larvae that were used in this work were obtained from colonies that were reared in the apiary at the College of Animal Sciences (College of Bee Science), Fujian Agriculture and Forestry University, Fuzhou City, China.

### 2.2. Source of sRNA-seq Data

The larval guts of *A. m. ligustica* that were 4-, 5-, and 6-days-old (Am4, Am5, and Am6 groups) were previously prepared. Each of the aforementioned three groups contained three larval guts, and there were three biological replicas that were used in this experiment [22]. Am4, Am5, and Am6 groups were subjected to cDNA library construction and deep sequencing using small RNA-seq (sRNA-seq) technology where 38, 011, 613; 43, 967, 518 and 39, 523, 034 raw reads were produced, and after quality control, 32, 524, 933; 36, 113, 035 and 27, 691, 488 clean tags were gained, respectively. The Pearson correlation coefficients between three different biological replicas within each group were above 98.22% [23]. The raw data that were generated from sRNA-seq were deposited in the NCBI SRA database under the BioProject number: PRJNA408312.

### 2.3. Identification and Investigation of piRNAs

*A. m. ligustia* piRNAs were identified according to our previously described protocol: (1) The clean reads were mapped to the reference genome of *A**. mellifera* (Assembly Amel_4.5), and the mapped clean reads were further aligned to GeneBank and Rfam (11.0) databases to remove small ncRNA including rRNA, scRNA, snoRNA, snRNA, and tRNA (2) miRNAs were filtered out from the remaining clean reads; and (3) sRNAs with a length distribution from 24 nt to 33 nt were screened out based on the length characteristics of piRNAs, and only those that aligned to a unique position were retained as candidate piRNAs. Next, first base bias of piRNAs in each group was summarized on the basis of the prediction result.

### 2.4. Target Prediction and Analysis of DEpiRNAs

The expression level of each piRNA was normalized to tags per million (TPM) following the formula TPM = T × 10^6^/N (T denotes clean reads of piRNA, N denotes clean reads of total sRNA). The fold change of the expression level of each piRNA between two different groups was determined following the formula: (TPM in Am5)/(TPM in Am4) or (TPM in Am6)/(TPM in Am5). On basis of the standard of *p*-value ≤ 0.05 and |log_2_(Fold change)| ≥ 1, DEpiRNAs in “Am4 vs. Am5” and “Am5 vs. Am6” comparison groups were screened out. TargetFinder software was used to predict the target genes of DEpiRNAs [24]. The targets were aligned to the GO (https://www.geneontology.org/, (accessed on 7 October 2022)) and KEGG (https://www.genome.jp/kegg/, (accessed on 7 October 2022)) databases using the BLAST tool to obtain corresponding annotation.

### 2.5. Construction and Analysis of Regulatory Network of DEpiRNAs

The gut of insects including the honey-bee is not only a key organ for food digestion and nutrition absorption [25], but also a pivotal position for immune defense and host-pathogen interactions [26]. In addition, the developmental process of the bee gut was suggested to be accompanied with the development of immune and energy metabolism [22]. Therefore, DEpiRNAs that are relevant to immune and energy metabolism pathways as well as corresponding regulatory networks were further investigated in this current work. Following the KEGG pathway annotations, the target genes annotated in development-, immune-, and energy metabolism-associated signaling pathways were further surveyed, respectively, and the threshold for screening the targeted binding relationship was set as a binding free energy of less than −15 kcal/mol; the regulatory networks were constructed based on the targeting relationship between DEpiRNAs and genes, followed by visualization utilizing Cytoscape software [27] with default parameters.

### 2.6. Validation of DEpiRNAs by Stem-Loop RT-PCR

The total RNA from 4-, 5-, and 6-day-old *A. m. ligustica* larval guts were extracted using a FastPure^®^ Cell/Tissue Total RNA Isolation Kit V2 (Vazyme, Nanjing, China). The concentration and purity of RNA were checked with a Nanodrop 2000 spectrophotometer (Thermo Fisher, Waltham, MA, USA). A total of five DEpiRNAs were randomly selected for stem-loop RT-PCR validation, including four (piR-ame-1146560, piR-ame-1183555, piR-ame-387266, and piR-ame-856650) from the “Am4 vs. Am5” comparison group and one (piR-ame-592661) from the “Am5 vs. Am6” comparison group. Specific stem-loop primers and forward primers (F) as well as universal reverse primers (R) were designed using DNAMAN software and then synthesized by Sangon Biotech Co., Ltd. (Shanghai, China). According to the instructions of HiScript ^®^ 1st Strand cDNA Synthesis Kit, cDNA was synthesized by reverse transcription using stem-loop primers and used as templates for PCR of DEpiRNA. Reverse transcription was performed using a mixture of random primers and oligo (dT) primers, and the resulting cDNA were used as templates for PCR of the reference gene snRNA U6. The PCR system (20 μL) contained 1 μL of diluted cDNA, 10 μL of PCR mix (Vazyme, Nanjing, China), 1 μL of forward primers, 1 μL of reverse primers, and 7 μL of diethyl pyrocarbonate (DEPC) water. The PCR was conducted on a T100 thermocycler (Bio-Rad, Hercules, CA, USA) under the following conditions: pre-denaturation step at 95 °C for 5 min; 40 amplification cycles of denaturation at 95 °C for 10 s, annealing at 55 °C for 30 s, and elongation at 72 °C for 1 min; followed by a final elongation step at 72 °C for 10 min. The amplification products were detected on 1.8% agarose gel electrophoresis with Genecolor (Gene-Bio, Shenzhen, China) staining.

### 2.7. Verification of DEpiRNAs by RT-qPCR

The RT-qPCR was carried out following the protocol of SYBR Green Dye kit (Vazyme, Nanjing, China). The reaction system (20 μL) included 1.3 μL of cDNA, 1 μL of forward primers, 1 μL of reverse primers, 6.7 μL of DEPC water, and 10 μL of SYBR Green Dye. RT-qPCR was conducted on an Applied Biosystems QuantStudio 3 system (Thermo Fisher, Waltham, MA, USA) following the conditions: pre-denaturation step at 95 °C for 5 min, 40 amplification cycles of denaturation at 95 °C for 10 s, annealing at 60 °C for 30 s, and elongation at 72 °C for 15 s, followed by a final elongation step at 72 °C for 10 min. The reaction was performed using an Applied Biosystems QuantStudio 3 Real-Time PCR System (Themo Fisher). All the reactions were performed in triplicate. The relative expression of piRNA was calculated using the 2^−ΔΔCt^ method [28]. Detailed information about the primers that were used in this work is shown in Appendix A.

### 2.8. Statistical Analysis

Statistical analyses were conducted with SPSS software (IBM, Amunque, NY, USA) and GraphPad Prism 7.0 software (GraphPad, San Diego, CA, USA). Data were presented as the mean ± standard deviation (SD). Statistics analysis was performed using Student’s *t*-test. Significant (*p* < 0.05) GO terms and KEGG pathways were filtered by performing Fisher’s exact test with R software 3.3.1 [29,30].

## 3. Results

### 3.1. Identification and Characterization of piRNAs in A. m. ligustica Larval Guts

A total of 843 piRNAs were identified in the larval guts of *A. m. ligustica*; among these, 764 piRNAs were shared by Am4, Am5, and Am6 groups, while 11, 67, and one, respectively, were unique. Additionally, the first base of piRNAs in Am4, Am5, and Am6 groups had a C bias (Figure 1A). Further investigation showed that the length distribution of the identified piRNAs in Am4, Am5, and Am6 groups were from 24 nt to 33 nt (Figure 1B), similar to the findings in other animals such as the Mongolian horse and *Scylla paramamosain* [31,32].

### 3.2. Differential Expression Profile of piRNA during the Developmental Process of Larval Guts

Here, 78 DEpiRNAs were identified in the “Am4 vs. Am5” comparison group, including 61 up-regulated and 17 down-regulated piRNAs. Among these, the most significantly up-regulated one was piR-ame-1009988 (log_2_FC = 14.52, *p* = 7.22 × 10^−5^), followed by piR-ame-14055 (log_2_FC = 14.52, *p* = 7.22 × 10^−5^) and piR-ame-456655 (log_2_FC = 14.52, *p* = 7.22 × 10^−5^), while the three most significantly down-regulated DEpiRNAs were piR-ame-1223398 (log_2_FC= −11.38, *p* = 3.19 × 10^−9^), piR-ame-1186994 (log_2_FC= −1.66, *p* = 0.041), and piR-ame-1077365 (log_2_FC= −1.65, *p* = 0.001) (Figure 2A). A total of seven DEpiRNAs were identified in the “Am5 vs. Am6” comparison group, including two up-regulated and five down-regulated piRNAs. Among these, the two most significantly up-regulated DEpiRNAs were piR-ame-1243913 (log_2_FC = 3.01, *p* = 0.019) and piR-ame-592661 (log_2_FC = 1.14, *p* = 0.005); whereas the most significantly down-regulated piRNAs were piR-ame-1173337 (log_2_FC = −10.96, *p* = 1.34 × 10^−5^) and piR-ame-31653 (log_2_FC = −10.96, *p* = 1.34 × 10^−5^), followed by piR-ame-1246710 (log_2_FC = −1.18, *p* = 0.045) (Figure 2B). Detailed information about DEpiRNAs is presented in Appendix A.

### 3.3. Target Prediction and Annotation of DEpiRNA

DEpiRNAs in the “Am4 vs. Am5” comparison group can target 9, 983 genes, which could be annotated to 20 biological process-related GO terms such as cellular processes and RNA biosynthetic processes, 11 molecular function-related GO terms such as cation channel activity and cation binding, and 19 cellular component-related GO terms such as cell and membrane parts (Figure 3A). DEpiRNAs in the “Am5 vs. Am6” comparison group can target 1, 936 genes, and these targets could be annotated to a total of 41 GO terms, including cation transport, cation channel activity, and cells. (Figure 3B).

In addition, the target genes of DEpiRNA in the “Am4 vs. Am5” comparison group could be annotated to 142 pathways such as Wnt signaling pathway, propanoate metabolism, and Hippo signaling pathway (Figure 4A). Those in the “Am5 vs. Am6” comparison group can be annotated to 101 pathways including the Hippo signaling pathway, RNA degradation, and phototransduction (Figure 4B).

### 3.4. Investigation of Regulatory Network between DEpiRNAs and Target Genes

In the “Am4 vs. Am5” comparison group, 54 up-regulated piRNAs could target 9, 398 genes, while 14 down-regulated piRNAs could target 3, 606 genes; each of these DEpiRNAs can target more than two genes, with piR-ame-742536 and piR-ame-856650 binding to the highest number of target genes (1, 421 and 1, 437). Additionally, two up-regulated piRNAs in the “Am5 vs. Am6” comparison group could target 604 genes, whereas four down-regulated piRNAs could target 1, 473 genes. Each of these DEpiRNAs can target more than two genes, with piR-ame-592661 and piR-ame-31653 linking to the highest number of target genes (447 and 839).

The regulatory network was constructed and analyzed, and the results showed that 202 and 58 target genes in the above-mentioned two comparison groups were involved in seven development-associated signaling pathways such as Hippo, Notch, and mTOR signaling pathways, whereas 255 and 39 targets were engaged in seven immune-associated pathways including endocytosis, the Jak/STAT signaling pathway, and ubiquitin-mediated proteolysis (Figure 5A). Additionally, 33 and 12 targets were found to be enriched in three energy metabolism pathways, namely sulfur metabolism, nitrogen metabolism, and oxidative phosphorylation (Figure 5B). Detailed information about the targeting relationship between DEpiRNAs and genes relative to the development, immune, and energy metabolism pathways are shown in Appendix A.

### 3.5. Stem-Loop RT-PCR and RT-qPCR Verification of DEpiRNA

The stem-loop RT-PCR results indicated that fragments with an expected size (about 60–80 bp) were amplified from five randomly selected five DEpiRNAs (Figure 6), which verified the expression of these DEpiRNAs in the *A. m. ligustica* larval gut.

Further, RT-qPCR results suggested that the expression trend of these five DEpiRNAs were consistent with sRNA-seq datasets, confirming the reliability of our transcriptome data (Figure 7).

## 4. Discussion

Here, sRNA-seq datasets derived from 4-, 5-, and 6-day-old *A. mellifera* larval guts were used following two major considerations: (1) the larval stage of the honey-bee lasts for 6 days, 1- and 2-day-old larvae are very small and manual transfer is likely to cause larval death. It was found after artificial transfer of 3-day-old bee larvae to 24-well culture plates that the larvae can maintain a high survival rate up to 6-days-old in a constant temperature and humidity chamber under lab conditions [33]. (2) we had already performed deep sequencing of 4-, 5-, and 6-day-old *A. m. ligustica* larval guts using sRNA-seq, and deciphered the differential expression profile of miRNAs and the putative roles of DEpiRNAs in the regulation of larval gut development based on the obtained high-quality sequencing data [23]. Our team previously predicted 596 piRNAs in the *A. m. ligustica* workers’ midguts based on bioinformatics [34]. In this current work, 843 piRNAs were identified in the larval guts of *A. m. ligustica*, with a length distribution among 24 nt~33 nt. Further analysis showed that as many as 519 (61.57%) piRNAs were shared by the workers’ midguts and larval guts of *A. m. ligustica*, whereas the numbers of unique piRNAs were 324 and 77. It is inferred that the shared piRNAs are likely to play a fundamental role in various developmental stages of *A. m. ligustica* larval guts, while the unique piRNAs may play different roles in different developmental stages. In view of the limited information on bee piRNA, the piRNAs that were identified in the present study further enrich the reservoir of piRNAs in European honey-bee and offer a valuable genetic resource for related studies on other bee species.

In animals, piRNAs were verified to participate in the regulation of growth, development, and embryogenesis. Based on the overexpression and knockdown of piRNA-3312, Guo et al. [35] found that piRNA-3312 targeted the gut esterase 1 gene to decrease the pyrethroid resistance of *Culex pipiens pallens*. Praher et al. [36] found that piRNAs were significantly differentially expressed in the early developmental stages of *Nematostella vectensis*, indicative of the regulatory role of piRNAs in development. Here, 78 and seven DEpiRNAs were observed in “Am4 vs. Am5” and “Am5 vs. Am6” comparison groups, respectively, indicating that the process of the larval gut of *A. m. ligustica* was accompanied by the differential expression of piRNAs, and these DEpiRNAs may be engaged in regulating development of *A. m. ligustica* larval gut. DEpiRNAs in the “Am4 vs. Am5” and “Am5 vs. Am6” comparison groups were found to target 9, 983 genes. In addition, 1, 936 target genes were involved in metal ion transport and calcium ion transport terms relative to biological processes, membrane parts, and membrane terms relative to cellular components, and cation channel activity and ion channel activity relative to molecular function. Additionally, the targets of DEpiRNAs in the aforementioned two comparison groups were involved in four and three development-associated terms such as metabolic processes and development processes and three and two immune-associated terms such as immune system processes and response to stimulus. Targets in these two comparison groups were engaged in 142 and 101 KEGG pathways, including fatty acid metabolism and propanoate metabolism relative to metabolism, mRNA surveillance pathway and RNA degradation relative to genetic information processing, and lysosomes and endocytosis relative to cellular processes. Further analysis indicated that the targets were engaged in seven and seven development-related pathways such as the Hippo signaling pathway and Wnt signaling pathway, seven and seven immune-related pathways such as endocytosis and the Jak/STAT signaling pathway, as well as three and three energy metabolism-related pathways such as nitrogen metabolism and sulfur metabolism. These results demonstrate that DEpiRNAs exerted a potential regulatory function in the *A. m. ligustica* larval guts by affecting many biological processes including development, immune defense, and energy metabolism.

Mondal et al. [37] confirmed that piRNA was capable of silencing gene expression in an siRNA-like manner. In the present study, a complex regulatory network between DEpiRNAs and target genes was observed, and all DEpiRNAs had more than two targets, implying that DEpiRNAs may be used by *A. m. ligustica* larvae to modulate target gene expression during gut development. Additionally, piR-ame-742536 (log_2_FC = 11.34, *p* = 5.42 × 10^−7^) and piR-ame-856650 (log_2_FC = −1.49, *p* = 0.0003) in the “Am4 vs. Am5” comparison group could target 1, 421 and 1, 437 genes, respectively, while piR-ame-592661 (log_2_FC = 1.14, *p* = 0.005) and piR-ame-31653 (log_2_FC = −10.96, *p* = 1.34 × 10^−5^) in the “Am5 vs. Am6” comparison group had 447 and 839 target genes, respectively. This indicated that the abovementioned four DEmiRNAs were likely to play crucial parts in the developmental process of *A. m. ligustica* larval guts.

As a highly conserved signaling pathway, the Wnt signaling pathway plays a key role in maintaining the development and homeostasis of animals and in promoting intestinal regeneration [38]. Shah et al. [39] discovered that silencing *Wnt-1* at the larval stage of *Tribolium castaneum* could result in larval death and abnormal pupal and adult development. Fu et al. [40] conducted knockout of the *HaWnt1* gene in *Helicoverpa armigera* using CRISPR/Cas9 technology, and the results showed that the HaWnt1 signaling pathway was essential for the embryonic development of *H. armigera*. Here, 68 DEpiRNAs in the “Am4 vs. Am5” comparison group could target 58 genes involving the Wnt signaling pathway, including piR-ame-247619 (log_2_FC = 2.01, *p* = 0.002), piR-ame-750627 (log_2_FC = 2.20, *p* = 0.003), and piR-ame-990954 (log_2_FC = 11.81, *p* = 5.82 × 10^−5^). Additionally, seven DEpiRNAs in the “Am5 vs. Am6” comparison group could target 15 Wnt signaling pathway-relevant genes, including piR-ame-1173337 (log_2_FC = −10.96, *p* = 1.34 × 10^−5^), piR-ame-31653 (log_2_FC = −10.96, *p* = 1.34 × 10^−5^), and piR-ame-260979 (log_2_FC = −11.20, *p* = 3.78 × 10^−5^). In insects, the Hippo signaling pathway participates in regulating body size as well as normal growth and development [41]. The *BmSd* gene was characterized as one of the Hippo signaling-pathway-related genes [42]. Yin et al. [42] detected that as many as 57.9% of *Bombyx mori* showed deformed wings after *BmSd* knockdown. Here, we found that 68 DEpiRNAs in “Am4 vs. Am5” group, such as piR-ame-456655 (log_2_FC = 14.52, *p* = 7.22 × 10^−5^), piR-ame-5 (log_2_FC = 14.52, *p* = 7.22 × 10^−5^), and 14055 (log_2_FC = 14.52, *p* = 7.22 × 10^−5^), could target 52 genes involving the Hippo signaling pathway, whereas five DEpiRNAs in “Am5 vs. Am6” group could target 22 genes involving the Hippo signaling pathway, including piR-ame-31653 (log_2_FC = −10.96, *p* = 1.34 × 10^−5^), piR-ame-1173337 (log_2_FC = −10.96, *p* = 1.34 × 10^−5^), piR-ame-260979 (log_2_FC = −11.20, *p* = 3.78 × 10^−5^), piR-ame-592661 (log_2_FC = 1.14, *p* = 0.005), and piR-ame-1246710 (log_2_FC = −1.18, *p* = 0.045). Together, these results suggested that corresponding DEpiRNAs may affect Wnt and Hippo signaling pathways through regulating target gene expression, further controlling the growth and development of *A. m. ligustica* larval guts. However, additional work is needed for the exploration of the underlying mechanism.

The immune system in insects is composed of the humoral immune system dominated by several signaling pathways such as Imd/Toll, JAK/STAT, JNK, and insulin, and the cellular immune system is represented by phagocytosis, melanization, autophagy, and apoptosis [43]. The JAK/STAT signaling pathway is not only implicated in regulating cell growth, differentiation, apoptosis, and inflammatory immunity but also participates in gut immunity via the modulation of intestinal stem cell proliferation and epithelial cell renewal [43,44]. Here, we observed that the JAK/STAT signaling-pathway-related genes were targeted by 54 DEpiRNAs (piR-ame-742536, piR-ame-1183555, and piR-ame-1233036, etc.) in the “Am5 vs. Am6” comparison group and one DEpiRNA (piR-ame-1243913) in the “Am5 vs. Am6” comparison group, suggestive of the involvement of corresponding DEpiRNAs in the regulation of immune defense in the larval guts. Tomato yellow leaf curlvirus has been proven to enter whitefly *Bemisia tabaci* midgut epithelial cells through receptor-mediated clathrin-dependent endocytosis [45]. Zhang et al. [46] reported that the inhibition of endocytosis induced the proliferation of *Drosophila* intestinal stem cells and massive gut hyperplasia, which further affected intestinal development and lifespan. In this study, endocytosis-associated genes were found to be targeted by 60 DEpiRNAs in the “Am4 vs. Am5” comparison group including piR-ame-14055 and piR-ame-456655, and five DEpiRNAs in the “Am5 vs. Am6” comparison group such as piR-ame-31653 and piR-ame-1173337, indicating that corresponding DEpiRNAs potentially regulated endocytosis during the development of larval guts. Together, these results demonstrated that DEpiRNAs as potential regulators participated in the development of *A. m. ligustica* larval guts. More efforts are required to elucidate the regulatory function of these DEpiRNAs. Several groups confirmed the feasibility and reliability of performing functional investigation of piRNAs following the technical platform similar to miRNAs, e.g., the expression and knockdown of a piRNA through feeding or injecting mimics and inhibitors [35,37]. In the near future, on the basis of findings in this work, we will further select key DEpiRNAs followed by expression and knockdown via feeding corresponding mimics and inhibitors to uncover their functions in the development of larval guts.

## 5. Conclusions

Taken together, 843 piRNAs were, for the first time, identified in the *A. m. ligustica* larval guts, and the first base of *A. m. ligustica* piRNAs had a C bias. A total of 78 piRNAs were differentially expressed in 5-day-old larval guts compared with 4-day-old larval guts, while only seven DEpiRNAs were detected in the 6-day-old larval gut compared with 5-day-old larval guts. Additionally, these DEpiRNAs could target 9, 983 and 1, 936 genes, respectively, which were engaged in 50 and 41 functional terms such as the developmental process and immune system process and 142 and 101 KEGG pathways such as the Wnt signaling pathway and endocytosis. Moreover, some DEpiRNAs may modulate the expression of corresponding target genes in the *A. m. ligustica* larval guts, further affecting sulfur metabolism, nitrogen metabolism, oxidative phosphorylation, endocytosis, and ubiquitin-mediated proteolysis as well as Hippo, Notch, mTOR, and Jak/STAT signaling pathways.

## Figures and Tables

**Figure 1 genes-13-01879-f001:**
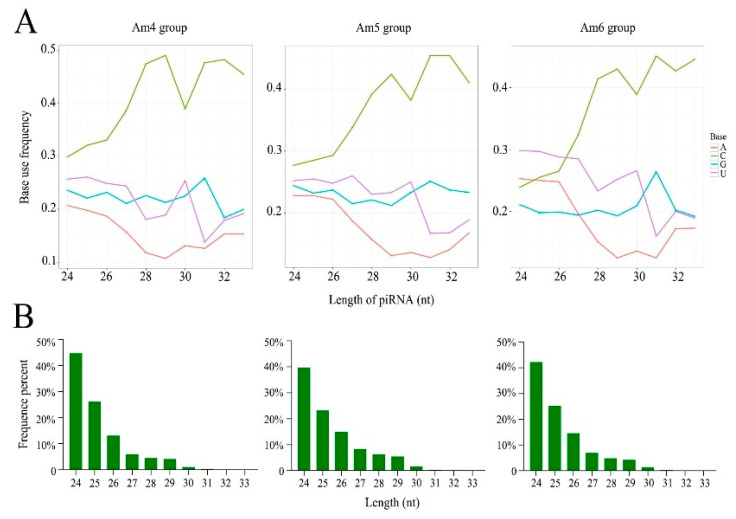
First base bias (**A**) and length distribution (**B**) of piRNAs that were identified in Am4, Am5, and Am6 groups.

**Figure 2 genes-13-01879-f002:**
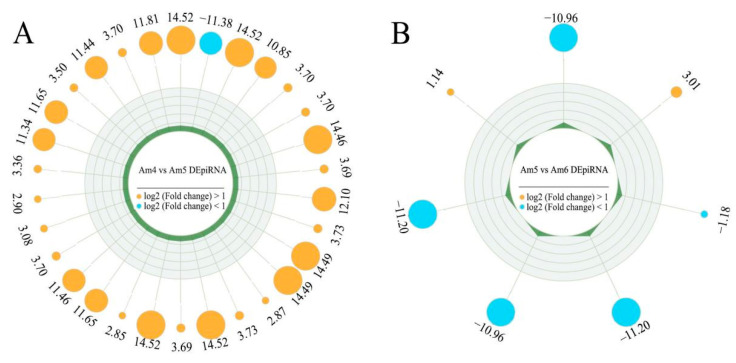
Radar maps of DEpiRNAs. (**A**) A total of 30 DEpiRNAs among 78 DEpiRNAs in the “Am4 vs. Am5” comparison groups. (**B**) A total of seven DEpiRNAs in the “Am5 vs. Am6” comparison groups.

**Figure 3 genes-13-01879-f003:**
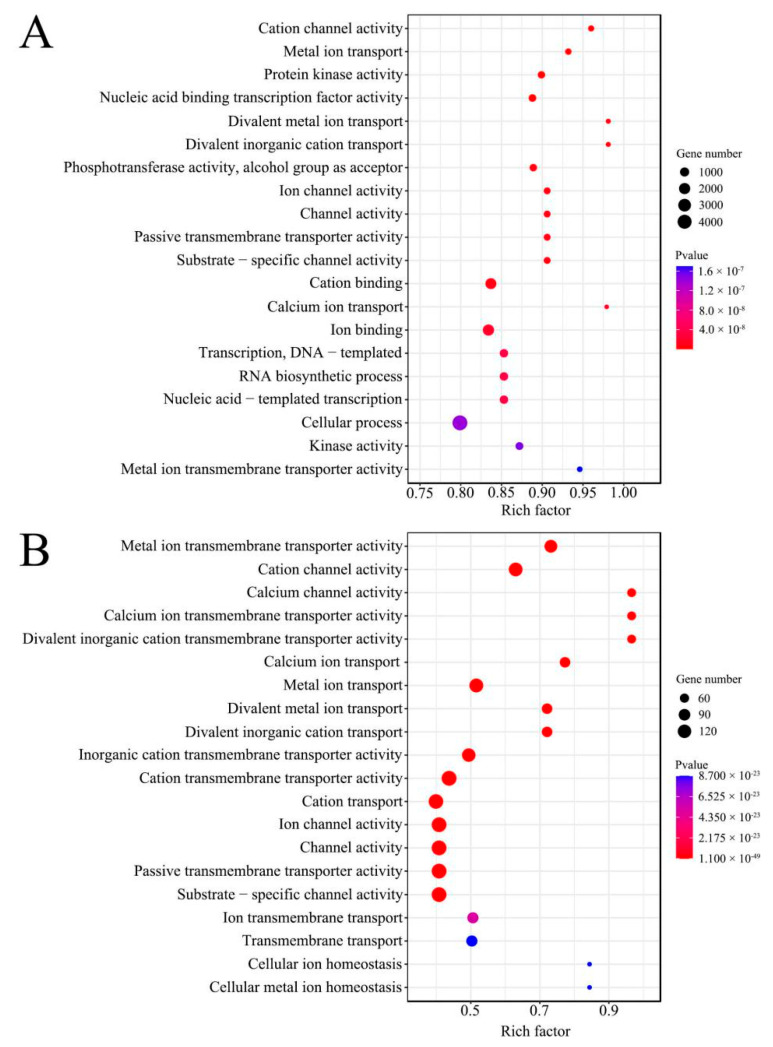
Top 20 GO terms that were annotated by target genes of DEpiRNAs in the “Am4 vs. Am5” (**A**) and “Am5 vs. Am6” (**B**) comparison groups. Circle size increases with the number of target genes, circle color depth increases significantly with *p*-value.

**Figure 4 genes-13-01879-f004:**
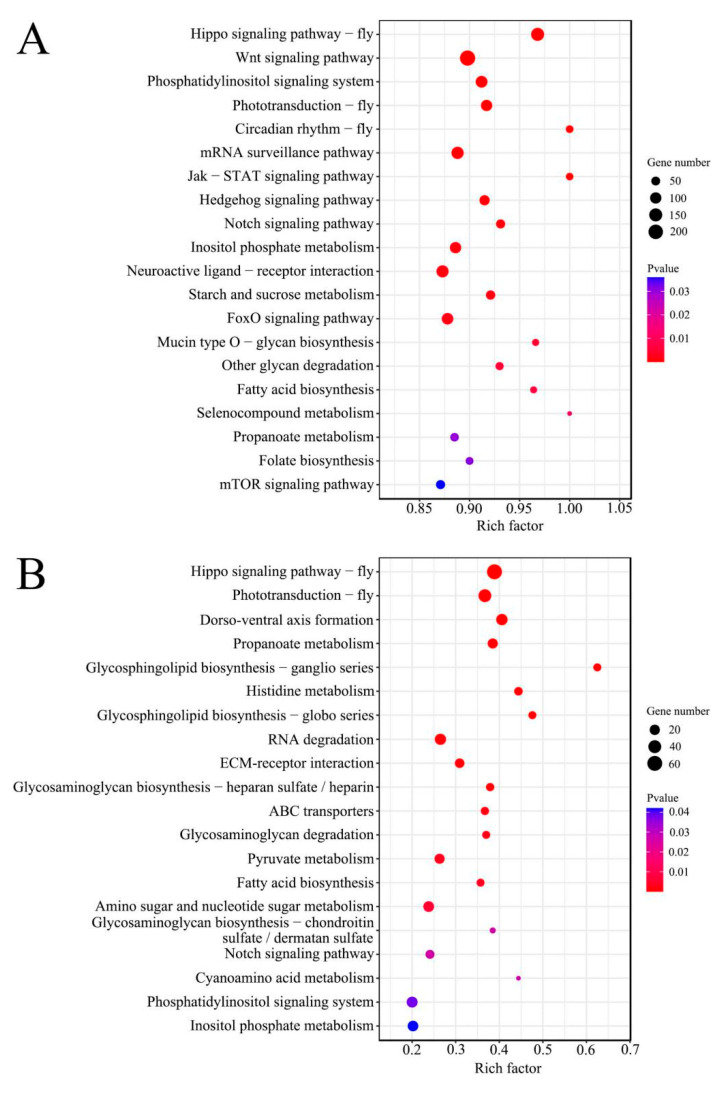
Top 20 KEGG pathways that were annotated by target genes of DEpiRNAs in the “Am4 vs. Am5” (**A**) and “Am5 vs. Am6” (**B**) comparison groups. Circle size increases with the number of target genes, circle color depth increases significantly with *p*-value.

**Figure 5 genes-13-01879-f005:**
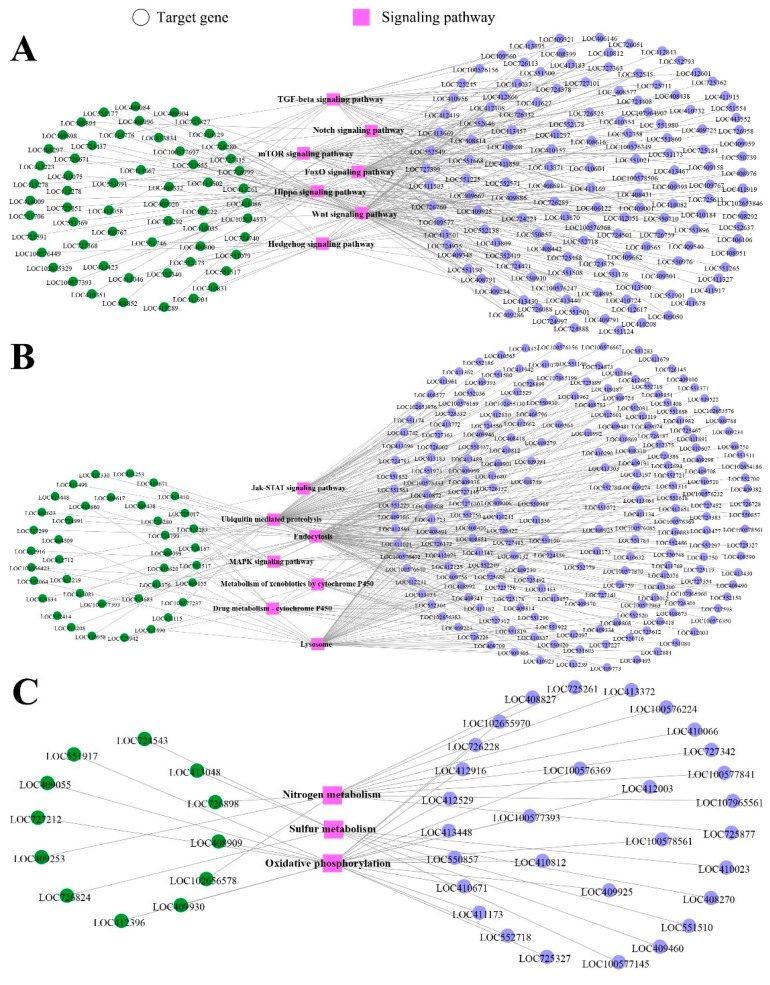
Regulatory networks of target genes of DEpiRNAs in the “Am4 vs. Am5” and “Am5 vs. Am6” comparison groups. (**A**) Regulatory network of DEpiRNAs and the corresponding targets relative to development-associated pathways. (**B**) Regulatory network of DEpiRNAs and the corresponding targets relative to immune-associated pathways. (**C**) Regulatory network of DEpiRNAs and the corresponding targets relative to energy metabolism-associated pathways. Only target gene ID and pathway names are presented while piRNA ID were omitted to make regulatory networks clearer. Purple circles indicate the target genes of DEpiRNAs in the “Am4 vs. Am5” comparison group, green circles indicate target genes of DEpiRNAs in the “Am5 vs. Am6” comparison group, while pink squares indicate signaling pathways.

**Figure 6 genes-13-01879-f006:**
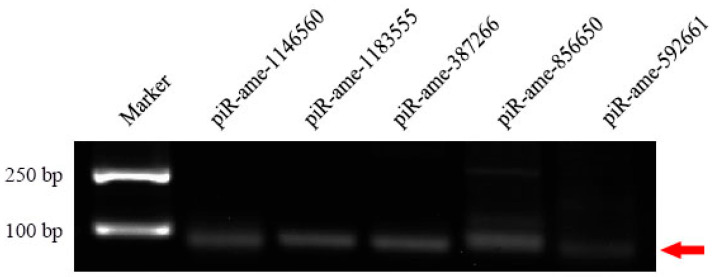
Agarose gel electrophoresis for amplification products from Stem-loop RT-PCR of five randomly selected DEpiRNAs. The red arrow indicates the signal band with the expected size (about 60–80 bp).

**Figure 7 genes-13-01879-f007:**
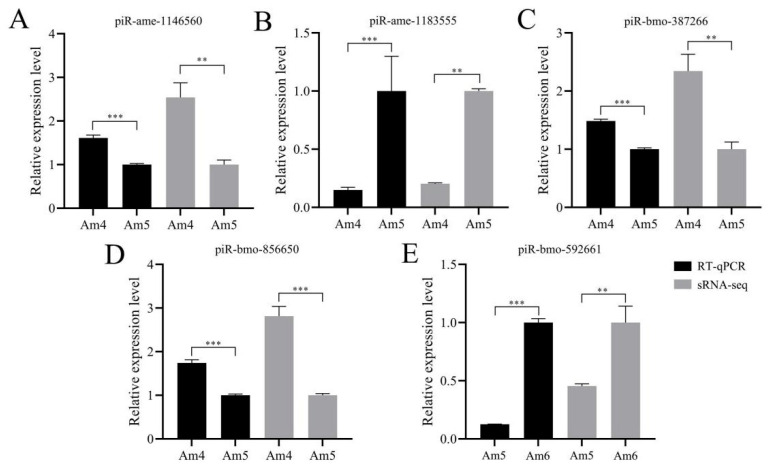
RT-qPCR detection of five randomly selected DEpiRNAs. The relative expression levels of the five DEpiRNAs that were detected by RT-qPCR were compared with those that were detected in sRNA-seq data. (**A**–**D**) Relative expression levels of piRNAs from the “Am4 vs. Am5” comparison group. (**E**) Relative expression level of piRNA from the “Am5 vs. Am6” comparison group. ** indicates *p* < 0.01 and *** indicates *p* < 0.001.

## Data Availability

Not applicable.

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
