# Peer review of "Transcriptome-Wide Characterization of piRNAs during the Developmental Process of European Honey-Bee Larval Guts"

_genes, 2022, doi:10.3390/genes13101879_

Round 1
Reviewer 1 Report
In this study, the authors investigated the mechanism underlying the development of European honey bee larval guts, from a new perspective of piRNA, a member of non-coding RNAs. In general, the work is interesting, the findings enrich the developmental biology of honeybees and are beneficial for those studying on insect gut development, and the manuscript is well written and easy to understand. However, several major and minor points should be addressed carefully.
Major points
1.The authors described that “piRNAs during the developmental process of European honey bee larval guts”, but only 4-, 5-, and 6-day-old larva guts were selected for investigation. You may need a few sentences to explain why these were selected.
2.I suggest adding a few sentences in introduction on non-coding RNA-regulated (eg. miRNA, lncRNA, circRNA) development of bee guts, to give more necessary information about the topic of this study.
3.Fruitful information about piRNAs expressed during the developmental process of honey bees were obtained on basis of bioinformatics, so what’s the significance of the big data, and what’s the next step to explore their real roles? New contents should be added into the Discussion section.
Minor points
1.Line 31: “were seven” is incomplete.
2.Line 47: add nt before 24, and maintain uniform throughout the whole manuscript.
3.Line 54-55: delete (1) and (2).
4.Line 64: replace “have a” with “exert”.
5.Line 68: change “pollination of” with “pollinating”.
6.Line 94: add “groups” before Am6.
7.Line 95: replace “with” with “using”.
8.Line 101: the BioProject number is missing.
9.Line 111: delete “piRNA”.
10.Line 122: delete “information”.
11.Line 123: what’s the reason of focusing on immune and energy metabolism?
12.Line 125: change “with” with “by”.
13.Line 127: I suggested the title was divided into two titles, one is Stem-loop RT-PCR and the other is RT-qPCR.
14.Line 145: the symbol of degrees Celsius is incorrect, please modify and maintain uniform.
15.Figure 1: Only a legend is needed.
16.Line 196: delete hyphen between biological and process.
17.add “” before comparison groups, check the whole manuscript.
18. figure 5. It is hard to understand. The title tells it is the regulatory network of DEpiRNAs, but I do not see DEpiRNAs in the figure. Please double check.
19.Line 255: bee>bees.
20.Line 261: add larval guts after A. m. ligustica.
21.Line 308: result>results.
22.Line 350: delete “it was regulated by”.
Author Response
We appreciate your comments and suggestions of great importance, which significantly improve the quality of our work and manuscript. Accordingly, we seriously checked and modified the manuscript, and all revision were showed in red in the revised version of manuscript.

Reviewer 2 Report
In the manuscript " Transcriptome-wide characterization of piRNAs during the developmental process of European honey bee larval guts ", Xu et al present a study in which they surveyed the piRNA population in three age groups of European honey bee larval guts, they discovered novel piRNAs and differentially expressed piRNAs comparing different age groups. They further predicted the potential gene target of these piRNAs and provided GO, KEGG and regulatory network analysis on the target genes.
I have several concerns as outlined below:
1. The authors include too many experimental details in the abstract, please be more concise.
2. Line 56 in Introduction section, “In the cytoplasm, piRNA precursors are transcribed from piRNA clusters” is not correct. piRNA precursors are transcribed from piRNA clusters in the nucleus.
3. In the Materials and Methods section, the authors should include more details about: (1) the number of replicates in each age group; (2) the method used for differential expression analysis of piRNAs; (3) the method used to construct the regulatory network.
4. piRNA precursors are single stranded, with no hairpin structure. How can the authors detect them using stem-loop RT-PCR or RT-qPCR? Please describe the rationale of the primer design and what kind of RNAs are detected, maybe with a simple schematic?
5. For the piRNAs identified, the authors can include more description of the features to further prove they are piRNAs. For example, size distribution, ping-pong signature.
6. The authors predicted the gene target of the piRNAs, but one important function of piRNA pathway is to repress TE activity. Do the piRNAs have any sequence similarity to the TEs?
7. There are 78 differentially expressed piRNAs in Am4 vs Am5 comparison, but only around 30 are plotted in Figure 2A. Please include the information about it in the figure legend.
8. In line 187, “whereas the most significantly down-regulated piRNAs was piR-ame-1246710 (log2FC=-1.18, P=0.045), followed by piR-ame-1173337 (log2FC=-10.96,
189 P=1.34E-05) and piR-ame-31653 (log2FC=-10.96, P=1.34E-05)” seems contradictory.
9. In figure 3, are all the GO terms significant? The authors should provide p-values of the GO terms.
10. In figure 4. The p-values for most the KEGG terms are not significant based on the figure legend. Please only include the significant terms that are meaningful.
11. In figure 5, is the regulatory network analysis between genes from Am4 vs Am5 comparison and between genes from Am5 vs Am6 comparison? Please explain the rationale underlining it.
12. In figure 7, the authors should include the error bars and p-values for the sRNA-seq results.
13. The authors should include more details in the text of all figure legends.
14. Please be more consistent of the way to write numbers in the text.
15. To provide some validation of the target gene prediction, the authors can do RT-qPCR of the target genes of the most differentially expressed piRNAs. If the piRNAs show a dramatic change, their target genes should in theory also show large corresponding changes.
Author Response

(The authors gave the same response as above.)

Round 2
Reviewer 2 Report
In the revised manuscript, the authors added new results and text, they also have ongoing work trying to validate the DEpiRNA target genes by RT-qPCR. They answered and addressed most of my concerns.
I have one more suggestion:
In line 52, the authors wrote “In somatic cells, piRNA precursors are transcribed from piRNA clusters in the nucleus…”. I think this primary piRNA pathway is actually true in both somatic cells and germline cells.
